# Modulation of GABA and resting state functional connectivity by transcranial direct current stimulation

Velicia Bachtiar[1], Jamie Near[2], Heidi Johansen-Berg[1], Charlotte J Stagg[1]*

[1]Oxford Centre for Functional MRI of the Brain, Nuffield Department of Clinical Neurosciences, University of Oxford, Oxford, United Kingdom; [2]Douglas Mental Health University Institute, Department of Psychiatry, McGill University, Montreal, Canada

**Abstract** We previously demonstrated that network level functional connectivity in the human brain could be related to levels of inhibition in a major network node at baseline (*Stagg et al., 2014*). In this study, we build upon this finding to directly investigate the effects of perturbing M1 GABA and resting state functional connectivity using transcranial direct current stimulation (tDCS), a neuromodulatory approach that has previously been demonstrated to modulate both metrics. FMRI data and GABA levels, as assessed by Magnetic Resonance Spectroscopy, were measured before and after 20 min of 1 mA anodal or sham tDCS. In line with previous studies, baseline GABA levels were negatively correlated with the strength of functional connectivity within the resting motor network. However, although we confirm the previously reported findings that anodal tDCS reduces GABA concentration and increases functional connectivity in the stimulated motor cortex; these changes are not correlated, suggesting they may be driven by distinct underlying mechanisms.

*For correspondence: charlotte. stagg@ndcn.ox.ac.uk

## Introduction

Recently, we demonstrated (*Stagg et al., 2014*) that the degree of connectivity within the motor resting state network (RSN) is negatively related to inhibition in the primary motor cortex (M1), a major network node. A similar relationship has also been established between GABA levels in the posteromedial cortex and the strength of the default mode network (*Kapogiannis et al., 2013*). However, while a relationship between resting functional connectivity and local inhibition has been established in the basal state, it has not yet been shown whether this relationship holds when local inhibition has been modulated, for example, after plasticity induction. RSNs are thought to reflect the brain's intrinsic functional architecture (*Beckmann et al., 2005*; *Smith et al., 2009*) and variation in the strength of these networks has been demonstrated to be modulated in a number of clinical conditions (*Westlake and Nagarajan, 2011*; *Woodward et al., 2014*; *Pievani et al., 2014*). However, their precise neurochemical basis has yet to be fully understood.

Both motor learning (*Floyer-Lea et al., 2006*) and anodal transcranial direct current stimulation (tDCS) applied to M1 (*Stagg et al., 2009, 2011a*) have previously been shown to decrease M1 GABA levels and to increase motor resting functional connectivity (*Albert et al., 2009*; *Sehm et al., 2012*; *Amadi et al., 2014*; *Stagg et al., 2014*). However, no study has yet directly investigated whether the increase in functional connectivity observed as a result of these interventions can be directly related to the concurrent GABA decreases.

Here, we aimed to extend our previous finding of a trait relationship between GABA levels in M1 and resting functional connectivity in a number of important ways. We wished to directly test the

relationship between the effects of tDCS on GABA and functional connectivity in the motor RSN. We wished to determine the changes in M1 GABA *during* anodal tDCS, which has not been previously measured. Additionally, we aimed to determine the duration of the previously reported GABA decrease after stimulation, which will have important implications for the design of future clinical studies.

## Results

12 healthy controls each attended two MR sessions, during which they received either anodal or sham tDCS (*Figure 1*). Magnetic resonance spectroscopy (MRS) data were acquired from the left M1 (*Figure 2A*) at baseline, during, and post either 20 min of 1 mA anodal tDCS applied to the left M1 or sham stimulation. Resting state fMRI data were acquired at baseline and post stimulation and were analysed using an independent component analysis approach (*Figure 2B*).

### Baseline GABA levels and M1 functional connectivity are inversely correlated

We first wished to investigate the previously described baseline relationship between GABA levels within M1 and the strength of motor functional connectivity in the same region. To ensure the specificity of this relationship, we calculated the baseline values within a 2 × 2 × 2 cm voxel corresponding to the location of the MRS acquisition for each individual separately. A partial correlation was then computed between the baseline GABA levels and baseline motor network strength within the MRS voxel across the two MR sessions, controlling for individual subjects. We demonstrated a significant inverse correlation between M1 GABA levels and the strength in motor functional connectivity within the MRS voxel (r(21) = −0.62, p < 0.01; *Figure 3*), though not within the whole motor network as a whole (r(21) = −0.23, p = 0.29).

### Anodal tDCS decreases GABA levels within M1

We then went on to investigate the time course of any M1 GABA changes induced by tDCS, using a repeated measures ANOVA with one factor of stimulation and one factor of time. As expected, there was no significant main effect of stimulation (F(1,11) = 3.22, p = 0.10) nor time (F(6,66) = 0.96, p = 0.46), but a significant stimulation × time interaction (F(6,66) = 2.29, p = 0.048; *Figure 4*). Post-hoc *t*-tests revealed significantly lower GABA post anodal tDCS compared with sham tDCS at both the post stimulation time points (paired *t*-test, Post1: t(11) = −4.14, p < 0.01; Post2: t(11) = −2.86, p = 0.02). GABA levels were also significantly lower post anodal tDCS compared with baseline (Post1: t(11) = −3.75, p < 0.01; Post2 t(11) = −2.51, p = 0.03) but not post sham tDCS compared with baseline (Post1: t(11) = 0.165, p = 0.87; Post2 t(11) = 0.75, p = 0.47). There were no significant differences between anodal and sham tDCS at any time point during stimulation (all p > 0.1).

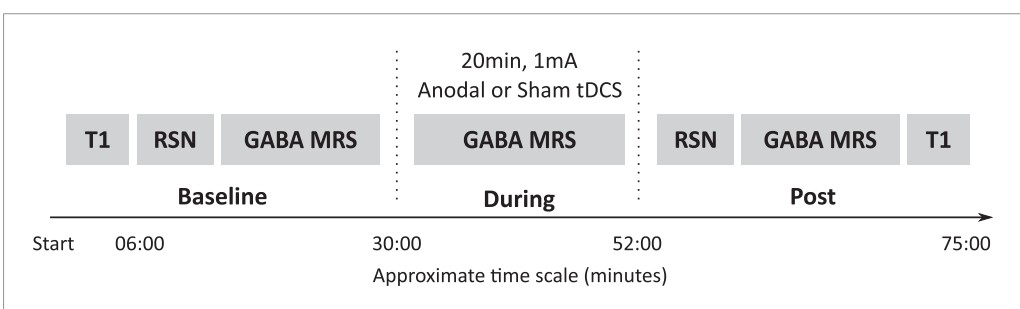

**Figure 1**. Experimental design. All subjects participated in two testing sessions with either anodal or sham tDCS, the order of which was counterbalanced across the group. GABA was measured at three time points (baseline, during, post) and resting state connectivity was measured at two time points (baseline, post). Timeline shown is an estimate of the length of the scans in minutes.

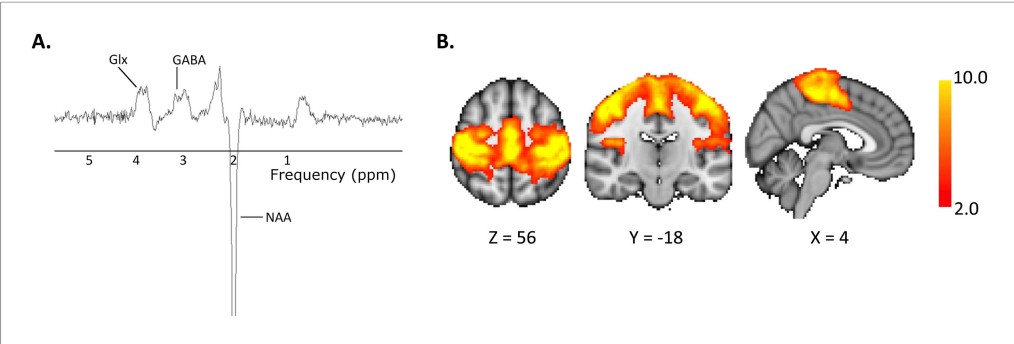

**Figure 2**. Representative (**A**) MR spectrum and (**B**) Group mean motor resting state network.

### Anodal tDCS increases functional connectivity within M1

To explore any tDCS-induced changes in motor network connectivity under the stimulating electrode, we wished first to define the region of M1 that showed the greatest connectivity with the rest of the motor network before stimulation. We therefore placed a 8 × 8 × 8 mm region of interest (ROI) centred on the peak coordinates of the motor RSN within the left M1. We calculated the mean motor network strength within this ROI for each subject, stimulation condition, and time point separately. As expected, a repeated measures ANOVA demonstrated no significant main effect of stimulation ($F(1,11) = 0.20$, $p = 0.67$) nor time ($F(1,11) = 2.78$, $p = 0.12$), but a significant stimulation × time interaction ($F(1,11) = 6.39$, $p = 0.03$; *Figure 5A*). Post-hoc *t*-tests demonstrated a significant increase in the strength of functional connectivity in the motor network within the left M1 ROI post anodal stimulation ($t(11) = -2.45$, $p = 0.03$), but not post sham stimulation ($t(11) = 0.07$, $p = 0.95$; *Figure 5B*).

### No relationship between change in GABA levels due to tDCS and change in functional connectivity

Finally, we wished to investigate whether the relationship between M1 GABA levels and M1 functional connectivity observed at baseline (*Figure 4*) held after modulation by tDCS. To do this, we compared change in GABA due to tDCS and the strength of motor network connectivity within the same voxel. There was no significant relationship between GABA levels post-tDCS [mean of Post1 and Post2] and functional connectivity within the MRS voxel post-tDCS, either when both anodal and sham stimulation were considered ($r = -0.27$, $p = 0.22$) or when anodal tDCS was considered alone ($r = -0.453$, $p = 0.14$; *Figure 6*). Neither was there any relationship between the tDCS-induced decrease in GABA levels (calculated as [(post average − baseline average)/baseline average]) and the tDCS-induced increase in functional connectivity (calculated as [(post − baseline)/baseline]) within the MRS voxel ($r = -0.21$, $p = 0.52$), nor within the motor network as a whole ($r = -0.01$, $p = 0.981$).

### Discussion

This study was performed to investigate the relationship between local inhibition within a major network node (here quantified via MRS-assessed GABA levels in M1) and functional connectivity within the motor network (quantified as network strength). In line with previous findings, we demonstrated that, at baseline, GABA levels within M1 correlate with the strength of the motor network at rest. We have also replicated the previous finding that anodal

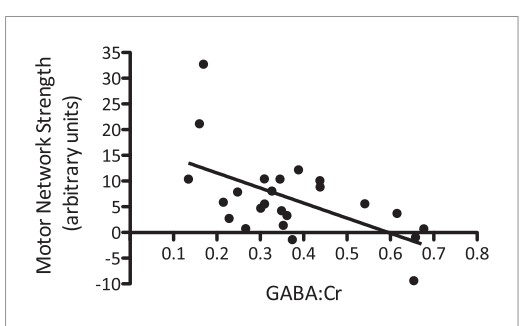

**Figure 3**. A significant relationship between M1-GABA and the strength of the motor network measured from the same region was identified at baseline ($r = -0.62$, $p < 0.01$).

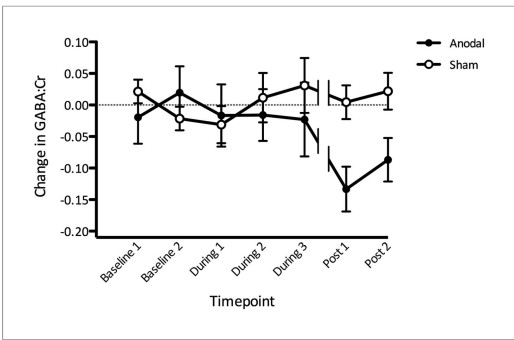

**Figure 4**. Change in GABA before, during, and after anodal tDCS relative to the sham condition. (**A**) Gradual decrease in GABA levels was observed during anodal stimulation with the most prominent decrease approximately 10–15 min after stimulation has finished (Post1). Break in the lines indicate a time gap when approximately 7 min of resting state acquisition was performed immediately after tDCS stimulation. Timescale shows the approximate time from the beginning of the scan session. (Post 1: t(11) = −4.14, p < 0.01; Post 2: t(11) = −2.86, p = 0.02).

tDCS reduces GABA concentration (*Stagg et al., 2009*, *2011a*) and increases functional connectivity in the stimulated cortex (*Sehm et al., 2012*; *Amadi et al., 2014*; *Stagg et al., 2014*). However, we show for the first time that the magnitude of these changes does not correlate across subjects, suggesting that they may be driven by distinct underlying mechanisms. We also provide novel evidence on the time course of GABA change with anodal tDCS and demonstrate that the previously described reduction in GABA is most prominent in the 30 min period after stimulation.

## Effects of anodal tDCS on GABA

As predicted, we saw a significant decrease in GABA levels in response to anodal tDCS with effects developing during stimulation and persisting for at least 30 min following stimulation. These results extend those of previous studies (*Stagg et al., 2009*, *2011a*), in which we were not able to acquire data during stimulation, and data were only acquired for 20 min post stimulation. As we could not practically continue beyond 30 min, it is not clear how long it would take for GABA to fully return to baseline after the stimulation period. Previous neurophysiological data show an increase in cortical excitability lasting for 90 min following 13 min of 1 mA anodal tDCS (*Nitsche and Paulus, 2001*), suggesting that much longer scan times may be required to fully track the time course of tDCS-induced GABA changes.

## Relationship between GABA and functional connectivity changes

To our knowledge, this is the first study to directly test for relationships between GABA and resting fMRI both at baseline and after plasticity induction. At baseline, in line with previous studies (*Kapogiannis et al., 2013*; *Stagg et al., 2014*), we found a significant negative correlation between the degree of resting inhibition in M1 and the degree of motor network functional connectivity. This finding reflects strong evidence from experimental and theoretical studies that long-range connectivity relates to local oscillatory activity in the beta and gamma bands (*Cabral et al., 2011*), activity which in turn is directly determined by local GABA_A synaptic activity (*Hall et al., 2011*).

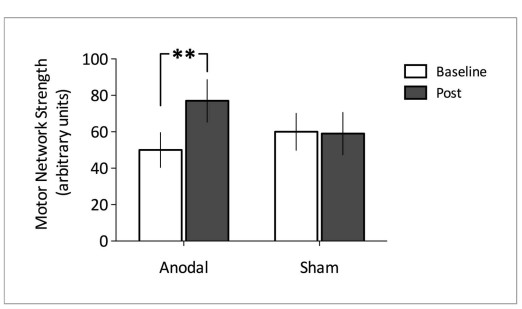

**Figure 5**. Change in functional connectivity before and after Anodal tDCS and the Sham condition. (**A**) Anodal tDCS applied to M1, significantly increased functional connectivity within M1 of the motor network (t (11) = −2.45, p = 0.03). (**B**) There were no differences in the sham condition (t(11) = 0.07, p = 0.95).

A number of studies have directly investigated the relationship between MRS-assessed GABA and oscillatory activity in the gamma range, with earlier studies demonstrating a direct relationship between the two measures (*Edden et al., 2009*; *Muthukumaraswamy et al., 2009*; *Gaetz et al., 2011*), findings that have not been replicated in a subsequent larger study (*Cousijn et al., 2014*). There may be a number of reasons for the lack of a consistent relationship between MRS-assessed GABA and gamma oscillations. GABA MRS is likely relatively insensitive to synaptic GABA_A activity, more closely reflecting extra-synaptic GABAergic tone (*Stagg et al., 2011b*). While at baseline functional GABA_A synaptic activity and GABAergic tone may be related, they are unlikely to be perfectly correlated, meaning that MRS-assessed GABA and gamma oscillations should

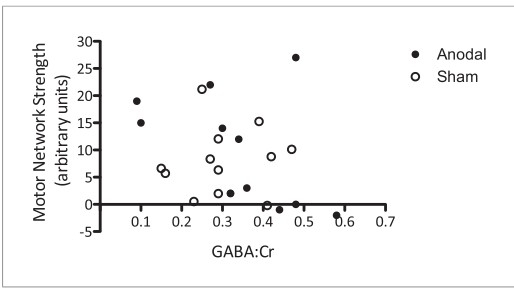

**Figure 6**. No significant relationship was demonstrated between GABA levels after tDCS and the strength of the motor network.

not necessarily be expected to be tightly related. In addition, although there is extensive data from animal models to demonstrate a clear relationship between beta and gamma oscillatory activity and $GABA_A$ activity, it is not necessarily clear how best to quantify oscillatory activity derived from MEG to most clearly demonstrate a similar relationship. It may be therefore, that differences in the acquisition or analysis of the MEG data mean that any relationship between gamma oscillations and GABA is difficult to reproduce using the techniques available in humans. Similarly, small differences in GABA MRS acquisition and analysis approaches may potentially impact on GABA estimates in ways that are not completely understood.

There was no significant relationship found between the tDCS-induced change in GABA levels and change in functional connectivity within the motor network.

There are a number of potential explanations for this lack of a relationship after plasticity induction, and we are unable to distinguish between these here. It may be that the time course of the tDCS-induced changes in GABA and functional connectivity are different, meaning that a direct comparison between functional connectivity immediately after stimulation and a later post-stimulation GABA measure would not demonstrate the expected relationship. In addition, as discussed in detail above, GABA MRS is likely relatively insensitive to synaptic $GABA_A$ activity, more closely reflecting extra-synaptic GABAergic tone (*Stagg et al., 2011b*). It may be therefore, that extra-synaptic GABA tone and $GABA_A$ synaptic activity are differentially modulated by tDCS, and hence MRS-assessed GABA is not a good surrogate marker for the change in oscillatory activity in the gamma and beta bands after stimulation, and hence long-range functional connectivity, which may be primarily dependent on $GABA_A$ synaptic activity. Alternatively, it may be that the relationship between GABA and functional connectivity is not causal and driven perhaps by another factor which we have not quantified. This seems unlikely given the evidence supporting the role of GABA in local oscillatory activity, and hence long-term connectivity but cannot be ruled out.

Given the close biochemical relationship between GABA and glutamate, GABA measures are unlikely to reflect overall changes in cortical excitability per se, and indeed no direct relationship between measures of GABA and cortical excitability has been demonstrated previously (*Stagg et al., 2011b*). The expected increases in cortical excitability following anodal tDCS have been shown to be multifactorial, and certainly are driven by modulation of both GABAergic and glutamatergic signalling (*Stagg and Nitsche, 2011*). It is worth noting, however, that the time course of GABA changes demonstrated here are broadly in line with those shown previously using paired pulse transcranial magnetic stimulation (ppTMS) (*Nitsche et al., 2005*). We did not acquire measures of excitability during or following tDCS in the subjects studied here. Future work could acquire excitability data over a similar time course to the one studied here using an interleaved TMS/MRI approach (*Siebner et al., 2003*), in order to test directly for any agreement between tDCS-induced changes in GABA and changes in excitability.

## Materials and methods

### Participants

Twelve healthy participants (four males; aged 22–28 years, mean 24 years) gave their informed consent to participate in this study in accordance with ethical approval from the East London Research Ethics Committee (Ref: 10/H0703/50). All participants were right handed as assessed by the Edinburgh Handedness Inventory (*Oldfield, 1971*).

### Experimental design

All subjects participated in two testing sessions with either anodal or sham tDCS. The sessions were separated by at least one week and the order of the sessions was counterbalanced across the group.

In each experimental session, GABA measurements were acquired at three time points: at baseline, during, and post-tDCS. Measures of resting state connectivity were acquired at baseline and immediately post-tDCS (*Figure 1*).

## tDCS

A DC-Stimulator (Magstim, Ltd; UK) delivered a 1-mA current to the brain via electrodes measuring 5 × 7 cm (Easycap, GmbH; Germany). One electrode was centred over the left M1 and positioned 5 cm lateral to mid-pre-central position (Cz). The reference electrode was placed over the contralateral supraorbital ridge. High Chloride Electrolyte-Gel (Easycap, GmbH; Germany) was used as the conducting medium between the scalp and electrodes. The electrodes contained 5 kΩ resistors and extension leads connected the stimulator, which was located outside of the magnetic field, to the subject positioned in the scanner. For anodal stimulation, the current was ramped up over 10 s and was then held at 1 mA for 20 min, before being ramped down over 10 s. For sham stimulation the DC-stimulator was ramped up for 10 s then switched off, as previously described (*Stagg et al., 2011a*).

## Magnetic resonance acquisition

All data were acquired on a 3-T Siemens Verio scanner. T1-weighted images (MPRAGE, 192 × 1 mm axial slices, TR/TE = 2040/5 ms, flip angle = 8°, FOV 200 × 181) were used to place a 2 × 2 × 2 cm voxel of interest over the left precentral knob, a known landmark for the hand motor representation (*Yousry et al., 1997*).

Edited GABA spectra were acquired using the MEGA-PRESS sequence (TR/TE = 2000/68 ms, 144 averages, TA = 4:56 blocks) with 20 ms double-banded Gaussian inversion pulses for simultaneous spectral editing and water suppression (*Mescher et al., 1998*). The water suppression band was set to a frequency of 4.7 ppm, and an editing band alternated between 1.9 ppm (edit on) and 7.5 ppm (edit off) in even and odd acquisitions, respectively. Spectra were acquired in blocks of 4.56 min. Three blocks were acquired at baseline, four blocks during tDCS, and three blocks post-tDCS.

A multiband 2 mm isotropic echo-planar imaging (EPI) acquisition (TR/TE = 1300/40 ms, FOV = 212 × 212 mm, bandwidth = 1814 Hz/Px, multi-band acceleration factor 6, voxel dimension = 2 mm isotropic, whole brain, acquisition time = 6:42 min for a total of 300 vol) (*Feinberg et al., 2010*; *Moeller et al., 2010*) was performed before and immediately after tDCS. Subjects fixated on a cross-hair image presented centrally on the screen during resting fMRI acquisition and watched a nature video at all other times.

## MRS analysis

MRS data pre-processing was performed using in-house scripts, including removal of motion-corrupted averages, frequency drift, and zero- and first-order phase corrections. To increase signal-to-noise, data from subsequent blocks within each time period were then averaged together, resulting in GABA measurements from 9:52 min acquisitions for the following time points: Baseline1, Baseline2, During1, During2, During3, Post1, and Post2.

MRS analysis was performed on the combined spectra using jMRUI v2.2 (http://www.mrui.uab.es/mrui/). As is standard, first, any residual water signal was removed using a Hankel Lanczos singular value decomposition (HLSVD) filter. Second, Creatine and NAA line-widths were obtained from the first non-edited acquisition for each time point using AMARES—a non-linear least square fitting algorithm (*Vanhamme et al., 2001*). The Creatine linewidth was used to constrain the linewidth of the GABA resonance from the edited spectra. The GABA resonance was fitted with 2 Gaussian peaks. A single Gaussian curve was fitted to the NAA resonance and was constrained to the linewidth of NAA in the non-edited spectrum. Spectra with an NAA linewidth of greater than 10 Hz were excluded from further analysis. Two spectra from different sessions in the same subject were excluded in this way.

FMRIB's automated segmentation tool (FAST), part of the FMRIB software library (http://www.fmrib.ox.ac.uk/fsl), calculated the relative quantities of grey matter and white matter within the voxel of interest on the T1-weighted structural scan as previously reported (*Stagg et al., 2011a*). As has been done previously, the amplitude of GABA peaks were corrected for the proportion of grey matter volume within the voxel (divided by [GM]/([GM] + [WM] + [CSF])), and Creatine concentrations were corrected for the proportion of total brain tissue volume within the voxel (divided by (([GM] + [WM])/

([GM] + [WM] + [CSF])) All neurotransmitter concentrations reported are given as a ratio to Creatine (GABA:Cr). Mean GABA:Cr values for each time point were normalised to the average of the baseline values for that condition, and henceforth are referred to as GABA levels for simplicity. We used Creatine as a simultaneously acquired reference peak as this is likely to remain stable over the timescale of this study. Note that in our original study (*Stagg et al., 2014*), this was not possible due to acquisition limitations with an older scanner, and so in that study we referenced GABA to NAA. However, we would not expect this to have a significant impact on our findings.

## Resting state functional connectivity analysis

Analysis of resting state fMRI data was carried out using an independent component analysis (ICA) approach as implemented in MELODIC (*Beckmann et al., 2005*) and tools from the FMRIB Software Library (FSL; http://www.fmrib.ox.ac.uk/fsl). Standard pre-processing steps were applied, which included motion correction, brain extraction, spatial smoothing using a Gaussian kernel of full-width at half-maximum (FWHM) of 2 mm, and high-pass temporal filtering equivalent to 100 s. fMRI volumes were registered to the individual's structural scan using boundary-based registration (BBR) (*Greve and Fischl, 2009*) and then to standard space images using FMRIB's nonlinear image registration tool (FNIRT). Pre-processed functional data containing 300 volumes for each time point and each subject were temporally concatenated across subjects to create a single 4D data set.

A dual regression technique was used to allow for between-subject voxel-wise comparisons of resting functional connectivity as previously described (*Filippini et al., 2009*; *Stagg et al., 2014*). To identify large-scale patterns of functional connectivity in the population of subjects, the concatenated 4D data set was decomposed using ICA into 25 components (*Stagg et al., 2014*). The motor RSN was identified manually by eye and then confirmed by using spatial cross-correlations against a previously defined motor map (*Beckmann et al., 2005*). A dual regression approach was then used to identify subject-specific temporal dynamics and associated spatial maps. This involved the following steps: for each subject the group-average set of spatial maps was regressed into the subject's 4D space-time data set. This generated a set of subject-specific time series for each group-level spatial map. Regressing these time series into the same 4D data set resulted in a set of subject-specific spatial maps, one for each group-level spatial map.

We then performed region of interest (ROI) analyses on the resulting subject-specific RSN map to investigate changes in motor resting state connectivity within specific ROIs. In all cases, the mean value within the ROI was extracted for each subject and used as a measure of the strength of functional connectivity within the RSN (*Stagg et al., 2014*).

To investigate the relationship between GABA levels and motor functional connectivity, individual subject masks of the MRS voxel were created to quantify the strength of motor network functional connectivity within the same region where the GABA was measured. In addition, to investigate the tDCS-induced changes in motor resting state connectivity, the voxel with the maximum degree of connectivity across the group was identified within the left precentral gyrus. This voxel was then dilated to create a 8 mm-radius ROI, which included the hand knob of M1 (*Yousry et al., 1997*). A mean measure of motor RSN functional connectivity strength within the ROIs was then extracted for each subject at each time point and condition separately.

## Statistical analysis

Statistical analyses on the data were conducted using SPSS (Version 22, IBM). Repeated measures ANOVAs (RM ANOVAs) were run with one factor of stimulation type (anodal, sham), and one factor of time (Baseline1, Baseline2, During1, During2, During3, Post1, Post2). Post-hoc Student's *t*-tests (paired samples, two-tailed) were performed as appropriate.

## Acknowledgements

VB is funded by the Clarendon Fund Scholarship. HJB is a Wellcome Trust Senior Research Fellow. CJS holds a Sir Henry Dale Fellowship jointly funded by the Wellcome Trust and Royal Society (Grant Number 102584/Z/13/Z). The research was supported by the National Institute for Health Research (NIHR) Oxford Biomedical Research Centre based at Oxford University Hospitals Trust, University of Oxford. The views expressed are those of the author(s) and not necessarily those of the NHS, the NIHR, or the Department of Health.

## Additional information

### Competing interests
HJ-B: Reviewing editor, *eLife*. The authors declare that no competing interests exist.

### Funding

| Funder | Grant reference | Author |
| --- | --- | --- |
| Wellcome Trust and Royal Society | 102584/Z/13/Z | Charlotte J Stagg |
| National Institute for Health Research (NIHR) | | Heidi Johansen-Berg, Charlotte J Stagg |
| Wellcome Trust | | Heidi Johansen-Berg |

The funders had no role in study design, data collection and interpretation, or the decision to submit the work for publication.

### Author contributions
VB, Conception and design, Acquisition of data, Analysis and interpretation of data, Drafting or revising the article, Contributed unpublished essential data or reagents; JN, Drafting or revising the article, Contributed unpublished essential data or reagents; HJ-B, CJS, Conception and design, Analysis and interpretation of data, Drafting or revising the article, Contributed unpublished essential data or reagents

### Ethics
Human subjects: Participants gave their informed consent to participate in this study in accordance with ethical approval from the East London Research Ethics Committee (Ref: 10/H0703/50).

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
