## [Decision Letter]

Thank you for submitting your work entitled “Modulation of GABA and resting state functional connectivity by transcranial direct current stimulation” for peer review at eLife. Your submission has been favorably evaluated by Eve Marder (Senior editor) and three reviewers of the original eLife manuscript (Stagg et al., eLife, 2014), one of whom, Jody Culham, is a member of our Board of Reviewing Editors.

The reviewers have discussed the reviews with one another and the Reviewing editor has drafted this decision to help you prepare a revised submission.

Summary:

This manuscript combines two approaches previously examined separately in an earlier eLife paper from a larger group in eLife (24). The earlier paper showed that (1) GABA levels in primary motor cortex (M1) measured by magnetic resonance spectroscopy (MRS) were inversely correlated with functional connectivity of M1 with other regions; and (2) anodal transcranial direct current stimulation (tDCS), a technique thought to reduce GABA levels, delivered to M1 led to an increase in its functional connectivity. Here the authors explicitly test the effects of tDCS to M1 on MRS GABA levels (before during and after tDCS) and functional connectivity. While they replicate both earlier findings, perhaps surprisingly, they don't find correlations between the degree of M1 GABA reductions and the increase in M1 connectivity.

The three reviewers agreed that the paper makes a valuable addition to the earlier results. Most notably, they investigate the cause and effect relationship between GABA levels and resting state fMRI connectivity and find that it is less straightforward than expected. Although these results may not be as exciting as one may have hoped, the reviewers were positive about the quality of the project and the Reviewing editor thought that this was a good example of how eLife Advances can contribute to a better understanding of earlier results (even if through adding caveats).

Essential revisions:

1) There was some confusion over the basis for the conclusion that post-tDCS GABA decreases have no relationship with connectivity measures. This warrants clarification.

One reviewer wrote, “Although the authors conclude no relationship given that GABA decreases significantly and connectivity increases significantly it would appear that there should be some correlation. I assume that the author's conclusion is based on the time course of connectivity being different (presumably the increases occur before the drop in GABA). However if this is the reasoning used it should be made clearer in the manuscript.”

The Reviewing editor assumed the basis was a near-zero correlation (subjects = data points) between them and recommends including a scatterplot to show this. As this seems to be the most important finding in this advance, it warrants its own figure (more so than some of the other data shown in figures, such as the fourth replication of the GABA:connectivity relationship).

2) As part of the study the authors reproduce their and other groups results which is commendable. However there has been some controversy recently on how well GABA changes track changes in cortical excitability (or gamma EEG) in other paradigms. The authors should briefly discuss these findings and offer an explanation for the discrepancy (methodological or biological) in results.

Recommended revisions/considerations:

3) Given that we do not know what constitutes a biologically significant drop in GABA, or increase in connectivity, I suggest that the authors also as a secondary analysis examine if there is a correlation if they take out the magnitude of the changes and just bin into increase, decrease, and no change. If no correlation is still seen then this analysis would bolster the author's conclusion.

4) Although the authors conclude that the impact of tDCS on GABA and resting state fMRI connectivity are independent I believe it is important that they also address the question of whether the drop in GABA has a role in the overall increase in cortical excitability. Given that they have now a time course of GABA drop I strongly suggest they examine whether there is a similar time course in the increase in cortical excitability in the region of the GABA voxel. If the two are in good agreement it would be important further evidence that GABA level has a modulatory role on cortical excitability and via that mechanism presumably plasticity.

---

## [Author Response]

*1) There was some confusion over the basis for the conclusion that post-tDCS GABA decreases have no relationship with connectivity measures. This warrants clarification*.

*One reviewer wrote, “Although the authors conclude no relationship given that GABA decreases significantly and connectivity increases significantly it would appear that there should be some correlation. I assume that the author's conclusion is based on the time course of connectivity being different (presumably the increases occur before the drop in GABA). However if this is the reasoning used it should be made clearer in the manuscript*.*”*

*The Reviewing editor assumed the basis was a near-zero correlation (subjects = data points) between them and recommends including a scatterplot to show this. As this seems to be the most important finding in this advance, it warrants its own figure (more so than some of the other data shown in figures, such as the fourth replication of the GABA:connectivity relationship)*.

We agree with the Reviewing editor that this is an important point and we are happy to clarify our discussion accordingly. It is difficult to know exactly why there is no correlation between the functional connectivity change and the GABA change – there are a number of potential explanations for this, and we are unable to distinguish between them here. We have expanded the discussion of this point in light of the reviewers’ comments, which we hope will clarify the justification for the conclusions for the reviewer:

“There are a number of potential explanations for this lack of a relationship after plasticity-induction, and we are unable to distinguish between these here. It may be that the time-course of the tDCS-induced changes in GABA and functional connectivity are different, meaning that a direct comparison between functional connectivity immediately after stimulation and a later post-stimulation GABA measure would not demonstrate the expected relationship. In addition, as discussed in detail above, GABA MRS is likely relatively insensitive to synaptic GABAA activity, more closely reflecting extra-synaptic GABAergic tone (27). It may be therefore, that extra-synaptic GABA tone and GABAA synaptic activity are differentially modulated by tDCS, and hence MRS-assessed GABA is not a good surrogate marker for the change in oscillatory activity in the gamma and beta bands after stimulation, and hence long-range functional connectivity, which may be primarily dependent on GABAA synaptic activity. Alternatively, it may be that the relationship between GABA and functional connectivity is not causal, and driven perhaps by another factor which we have not quantified. This seems unlikely given the evidence supporting the role of GABA in local oscillatory activity, and hence long-term connectivity, but cannot be ruled out.”

We are very happy to add in the scatter plot as the Reviewing editor suggests, and have now incorporated this into the manuscript (Figure 6).

*2) As part of the study the authors reproduce their and other groups results which is commendable. However there has been some controversy recently on how well GABA changes track changes in cortical excitability (or gamma EEG) in other paradigms. The authors should briefly discuss these findings and offer an explanation for the discrepancy (methodological or biological) in results*.

We are happy to include a discussion of this point:

“A number of studies have directly investigated the relationship between MRS-assessed GABA and oscillatory activity in the gamma range, with earlier studies demonstrating a direct relationship between the two measures (16; 6; 10), findings that have not been replicated in a subsequent larger study (5). There may be a number of reasons for the lack of a consistent relationship between MRS-assessed GABA and gamma oscillations. GABA MRS is likely relatively insensitive to synaptic GABAA activity, more closely reflecting extra-synaptic GABAergic tone (27). While at baseline functional GABAA synaptic activity and GABAergic tone may be related, they are unlikely to be perfectly correlated, meaning that MRS-assessed GABA and gamma oscillations should not necessarily be expected to be tightly related. In addition, although there is extensive data from animal models to demonstrate a clear relationship between beta and gamma oscillatory activity and GABAA activity it is not necessarily clear how best to quantify oscillatory activity derived from MEG to most clearly demonstrate a similar relationship. It may be therefore, that differences in the acquisition or analysis of the MEG data mean that any relationship between gamma oscillations and GABA is difficult to reproduce using the techniques available in humans. Similarly, small differences in GABA MRS acquisition and analysis approaches may potentially impact on GABA estimates in ways that are not completely understood.”

Recommended revisions/considerations:

*3) Given that we do not know what constitutes a biologically significant drop in GABA, or increase in connectivity, I suggest that the authors also as a secondary analysis examine if there is a correlation if they take out the magnitude of the changes and just bin into increase, decrease, and no change. If no correlation is still seen then this analysis would bolster the author's conclusion*.

We thank the reviewers for this suggestion. We have performed the analysis as requested, and have binned both the change in GABA and change in functional connectivity into “increase” and “decrease” (No subject demonstrated no change in either measure). Analysis of this data showed no significant correlation between these two bins, either when all the post-stimulation data was considered (χ^2^ = 0.167, p = 0.68), or when the anodal tDCS data was considered alone (χ^2^ = 1.33, p = 0.25). We have not added this analysis to the manuscript, but are happy to do so if the reviewers and editor feel it is appropriate.

*4) Although the authors conclude that the impact of tDCS on GABA and resting state fMRI connectivity are independent I believe it is important that they also address the question of whether the drop in GABA has a role in the overall increase in cortical excitability. Given that they have now a time course of GABA drop I strongly suggest they examine whether there is a similar time course in the increase in cortical excitability in the region of the GABA voxel. If the two are in good agreement it would be important further evidence that GABA level has a modulatory role on cortical excitability and via that mechanism presumably plasticity*.

This is an interesting point, but it is difficult to know how to best address this question directly. As demonstrated previously (Nitsche et al., J Physiol, 2005), substantially different mechanisms underlie the cortical excitability changes during and after anodal tDCS, findings which would be broadly in line with the results described in our paper.

In their 2005 paper, Nitsche and colleagues demonstrated a trend towards an increase in cortical excitability during anodal tDCS but with no change in SICI, the TMS measure of GABAA activity. This would be broadly in line with our findings here, where there was no significant decrease in MRS-assessed GABA until after the stimulation period had ceased.

Conversely, 13 minutes of 1mA tDCS has been shown to lead to an increase in cortical excitability that outlasts the stimulation period by 60-90 minutes (Nitsche & Paulus, Neurology, 2001; Nitsche et al., J Physiol, 2005; Monte-Silva et al., Brain Stimulation, 2012). This increase in cortical excitability has been demonstrated to be broadly stable from the offset of stimulation for at least 60 minutes, in line with our GABA findings here. This increase in cortical excitability occurs concurrently with both an increase in intracortical facilitation (ICF; a TMS measure of both glutamate and GABA) and a decrease in SICI, suggesting that the increase in cortical excitability seen after anodal tDCS is driven by both a decrease in GABA and an increase in glutamatergic activity (Nitsche et al., J Physiol, 2005). It is important to note that although 20 minutes of 1mA anodal tDCS has been shown to be excitatory to M1 (O’Shea et al., NeuroImage, 2014) the timecourse of cortical excitability changes has not directly been studied to our knowledge and no ppTMS measures performed.

We do not have TMS measures of cortical excitability in the subjects in the current paper, but we would not wish to argue that the expected increase in excitability was driven solely by GABA changes, or indeed that the magnitude of GABA changes and the degree of overall excitability increase were necessarily related in an individual subject. As stated above, the increase in cortical excitability is almost certainly multi-factorial and driven at least by increases in glutamatergic signaling as well as decreases in GABA. MRS, providing as it does concentration measures, does not given a good measure of glutamatergic activity and therefore is not in our opinion the best modality to assess this question.

Rather, we would like to argue that while increases in excitability are driven by multifactorial effects, changes in GABA per se can be related to one aspect of these – long range functional connectivity. We have added the following to the discussion to clarify this point:

“Given the close biochemical relationship between GABA and glutamate, GABA measures are unlikely to reflect overall changes in cortical excitability per se, and indeed no direct relationship between measures of GABA and cortical excitability has been demonstrated previously (27). The expected increases in cortical excitability following anodal tDCS have been shown to be multifactorial, and are certainly driven by modulation of both GABAergic and glutamatergic signalling (28). It is worth noting, however, that the timecourse of GABA changes demonstrated here are broadly in line with those shown previously using paired pulse transcranial magnetic stimulation (ppTMS) (18). It is worth noting, however, that the timecourse of GABA changes demonstrated here are broadly in line with those shown previously using paired pulse transcranial magnetic stimulation (ppTMS) (18). We did not acquire measures of excitability during or following tDCS in the subjects studied here. Future work could acquire excitability data over a similar timecourse to the one studied here using an interleaved TMS/MRI approach (22), in order to test directly for any agreement between tDCS induced changes in GABA and changes in excitability.”